# Impact of San Francisco's New Street crisis response Team on Service use among people experiencing homelessness with mental and substance use disorders: A mixed methods study protocol

Matthew L. Goldman[1,2]*, Megan McDaniel[1], Deepa Manjanatha[1,3], Monica L. Rose[1], Glenn-Milo Santos[1,4,5], Starley B. Shade[4,6], Ann A. Lazar[4,7], Janet J. Myers[8,9], Margaret A. Handley[4,8,9], Phillip O. Coffin[1,8]

1 San Francisco Department of Public Health, San Francisco, San Francisco, CA, United States of America, 2 Department of Psychiatry and Behavioral Sciences, University of California, San Francisco, San Francisco, CA, United States of America, 3 San Diego State University/University of California, San Diego Joint Doctoral Program in Clinical Psychology, San Diego, CA, United States of America, 4 Department of Epidemiology and Biostatistics, University of California, San Francisco, San Francisco, CA, United States of America, 5 Department of Community Health Systems, University of California, San Francisco, San Francisco, CA, United States of America, 6 Institute for Global Health Sciences, University of California, San Francisco, San Francisco, CA, United States of America, 7 Division of Oral Epidemiology, University of California, San Francisco, San Francisco, CA, United States of America, 8 Department of Medicine, University of California, San Francisco, San Francisco, CA, United States of America, 9 UCSF Partnerships for Research in Implementation Science for Equity (PRISE Center), San Francisco, San Francisco, CA, United States of America

* matthew.goldman@ucsf.edu

**Data Availability Statement:** This manuscript is for a study protocol in which we don't have data or results to potentially share with a third party.

## Abstract

Mobile crisis services for people experiencing distress related to mental health or substance use are expanding rapidly across the US, yet there is little evidence to support these specific models of care. These new programs present a unique opportunity to expand the literature by utilizing implementation science methods to inform the future design of crisis systems. This mixed methods study will examine the effectiveness and acceptability of the Street Crisis Response Team (SCRT), a new 911-dispatched multidisciplinary mobile crisis intervention piloted in San Francisco, California. First, using quantitative data from electronic health records, we will conduct an interrupted time series analysis to quantitatively examine the impacts of the SCRT on people experiencing homelessness who utilized public behavioral health crisis services in San Francisco between November 2019 and August 2022, across four main outcomes within 30 days of the crisis episode: routine care utilization, crisis care reutilization, assessment for housing services, and jail entry. Second, to understand its impact on health equity, we will analyze racial and ethnic disparities in these outcomes prior to and after implementation of the SCRT. For the qualitative component, we will conduct semi-structured interviews with recipients of the SCRT's services to understand their experiences of the intervention and to identify how the SCRT influenced their health-related trajectories after the crisis encounter. Once complete, the quantitative and qualitative findings will be further analyzed in tandem to assist with more nuanced understanding of the

**Funding:** This work was supported by the Robert Wood Johnson Foundation's Health Systems Transformation Research Coordinating Center Call for Proposals: Research to Advance Models of Care for Medicaid-Eligible Populations (Grant #78236). The funders had no role in study design, data collection and analysis, decision to publish, or preparation of the manuscript.

**Competing interests:** Dr. Goldman is a paid research consultant for Vibrant Emotional Health, the National Council for Mental Wellbeing, Peg's Foundation, the University of California, Davis, and the Research Foundation for Mental Hygiene, Inc. This does not alter our adherence to PLOS ONE policies on sharing data and materials.

effectiveness of the SCRT program. This evaluation of a novel mobile crisis response program will advance the field, while also providing a model for how real-world program implementation can be achieved in crisis service settings.

## Introduction

Mobile crisis services for people experiencing distress related to mental health or substance use are expanding rapidly across the US [1]. Mobile crisis has a unique ability to respond rapidly in a less restrictive environment [2] and to coordinate with community partners such as law enforcement and emergency departments to divert people from those settings [3]. Recent federal legislation incentivized Medicaid coverage for mobile crisis services, and state and local governments have begun to invest significantly in expanding these programs [4].

With suicide rates and overdose deaths continuing to climb [5, 6], scarce resources and strained workforce must be positioned to be as high impact as possible. Although clinical trials in crisis services are often unfeasible given the high acuity of clinical scenarios and limitations to ethical randomization to experimental conditions, the creation of a range of crisis programs in real-world settings presents an opportunity to use implementation science methods to characterize which programs and models are meeting their stated objectives and informing future best practices [7].

Prior single-site quasi-experimental studies of mobile crisis programs have found impacts on service utilization and costs [8]. However, there are many remaining questions about how effective mobile crisis teams are at linking people to routine care and social services or at preventing adverse outcomes such as jail entry or reutilization of acute care services. Furthermore, while mobile crisis programs are often justified by reducing criminalization of people with mental illness, few studies have focused on programs that target high-risk populations such as people experiencing homelessness (PEH) [9, 10].

This paper describes the research protocol for an evaluation of a Street Crisis Response Team (SCRT) in San Francisco, California, a model that tailors its services to PEH. Mental illness and substance use disorders are highly prevalent among adult PEH in San Francisco, yet access to appropriately tailored services is limited. Especially troubling is the inequity of the burden of these diagnoses within this population: a third of PEH in San Francisco identify as Black/African American, compared to 5% of the overall population. Across the US, people with serious mental illness comprise approximately one quarter of all PEH, and up to one third has a substance use disorder [2], with people of color dramatically over-represented in this population [11]. Despite these trends, engaging PEH in mental health and substance use care as well as social services is impeded by marginalization, dehumanization, and structural violence, which interfere with trust and engagement in health care and social services [12].

We will employ an implementation science approach to study a novel mobile crisis program by drawing on empirical data from the health care system as well as perspectives from service recipients, which will allow for a deeper understanding of the utility—and potential limitations—of measuring traditional service outcomes in this setting [13]. We anticipate that this study will yield findings that inform both the implementation of existing, and the planning and evaluation of future mobile crisis programs.

## Methods

To evaluate the impact of the SCRT, we will use a QUANT-QUAL mixed methods implementation science approach [14, 15]. First, we will use quantitative methods to examine if there are

changes in utilization of mental health, substance use, and housing services as well as jail entry following implementation of the SCRT among PEH who present in behavioral health crisis to acute care settings within San Francisco's public health system. Second, we will evaluate the ability of the SCRT to enhance equity by stratifying our analysis by ethnoracial groups to examine pre-implementation disparities as well as post-implementation worsening, perpetuation, or resolution of baseline disparities. We will use our quantitative analysis to set the sampling frame for qualitative semi-structured interviews, which will be conducted with the SCRT service recipients to understand the facilitators and barriers to achieving their goals. Finally, we will consider the quantitative and qualitative results in combination to help interpret both sets of findings.

Study procedures were approved by the University of California, San Francisco, Committee on Human Subjects Research (Protocol #20–32693). The Committee waived consent for review of healthcare, housing, and jail records, and approved verbal informed consent procedures for telephonic and in-person interview participants.

## Program

The Behavioral Health Services division of the San Francisco Department of Public Health (SFDPH) has an extensive infrastructure for mental health and substance use disorder services, yet one important gap has remained: real-time response for people in behavioral health crisis in the streets. In 2019, San Francisco's 9-1-1 call data indicates that approximately 50,000 behavioral health related calls were received, most of which were responded to by a law enforcement unit. Because most calls were not related to criminal events, and instead to mental health and social needs experienced by PEH, San Francisco behavioral health and governmental leaders worked with community stakeholders to create the SCRT to respond as an alternative able to be more responsive to behavioral health crises that happen on the street [16].

The SCRT was designed based on previous co-responder models [17, 18]. To meet the goal of diverting calls that would typically go to the San Francisco Police Department, this trauma-informed specialty behavioral health team is dispatched solely by 9-1-1 operators. The SCRT utilizes a co-responder model comprised of a behavioral health clinician, a paramedic from the San Francisco Fire Department and a peer specialist. Each team member plays a role in providing care including immediate stabilization of urgent medical need (paramedic), de-escalation of the crisis (behavioral health clinician) and person-centered peer support (peer specialist). The team triages clients to the appropriate level of care, be it through resolution of the crisis in the field, linkage to outpatient mental health and substance use services, or transport to an acute treatment setting. The SCRT was first piloted in San Francisco's highest demand neighborhoods in December 2020 and then was incrementally expanded to be citywide by June 2021 (Fig 1).

The SCRT also includes follow-up services provided by the Office of Coordinated Care (OCC), which is charged with offering support after the SCRT encounter with the goal of linking clients to outpatient mental health and housing services, thus reducing reutilization of acute services.

## Quantitative methods

To assess the implementation of the SCRT, we will use an interrupted time series (ITS) design, which is a quasi-experimental method that allows for non-randomized evaluation of an intervention [19]. In this ITS study, we will measure the effect of the intervention by generating models to assess changes over time in each outcome before and after implementation of the SCRT. This will allow us to model the secular trends in data not due to the intervention itself.

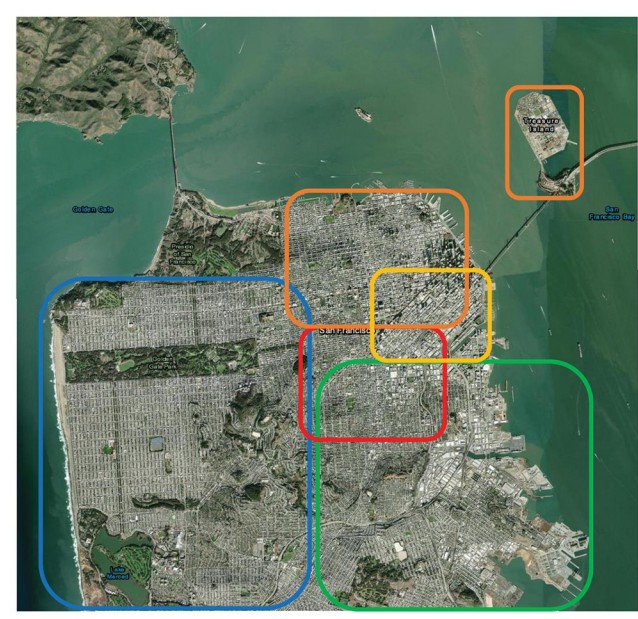

### Coverage & Hours

| Region | Hours | Launch Date |
|---|---|---|
| Tenderloin | 0900-2100 | 11/30/2020 |
| Mission/Castro | 0700-1900 | 2/1/2021 |
| Bayview | 1100-2300 | 4/5/2021 |
| Waterfront/Chinatown/North Beach | 0700-1900 | 5/10/2021 |
| Park/Richmond/Sunset | 0600-1800 | 6/14/2021 |
| Citywide Overnight | 1830-0630 | 7/26/21 |
| Citywide Overnight | 1900-0700 | 5/28/22 |

Landsat-7 image courtesy of the U.S. Geological Survey.

**Fig 1. Rollout of the street crisis response team across San Francisco by region, hours of operation, and launch date of each phase of expansion.** Landsat-7 image courtesy of the U.S. Geological Survey.

**Study population.** The SCRT aims to provide specialty mental health response and enhanced resources to adults in San Francisco who experience crises related to mental health and/or substance use disorders and are experiencing homelessness. Therefore, the study population will be defined based on age greater than 18 years, meeting criteria for homelessness in the 12 months prior to or 3 months following the crisis episode, and receipt of crisis care from any of San Francisco's "front door" programs for people in an acute behavioral health crisis. These settings include two mobile crisis programs (Comprehensive Crisis Services and the SCRT), a crisis stabilization unit (DORE Urgent Care Clinic), and emergency psychiatric services (Zuckerberg San Francisco General Hospital's Psychiatric Emergency Services [PES] and Emergency Department visits with a primary behavioral health diagnosis) (Fig 2). The population of housed adults utilizing acute behavioral health services will be utilized as a control group in the ITS sensitivity analyses.

**Data sources and matching procedures.** The primary data sources for our quantitative analysis will include electronic health record (EHR) data from the network of clinics funded by the city's health plan, public housing assessment data, and jail entry data. We will integrate data from SFDPH's two main EHR vendors, Avatar (NetSmart) and EPIC, which are used by SFDPH mental health and substance use treatment providers, medical clinics, mobile crisis teams, crisis stabilization units, and Zuckerberg San Francisco General Hospital's medical ED, psychiatric emergency services (PES) and inpatient psychiatry. Homelessness and housing assessment data originate from the Homeless Management Information System (HMIS), which is used by all entry points into the housing service system in San Francisco and automatically links data into the EPIC EHR. Jail entry data originates from the Jail Information Management System (JIMS), which the Department of Public Health's Jail Health Services

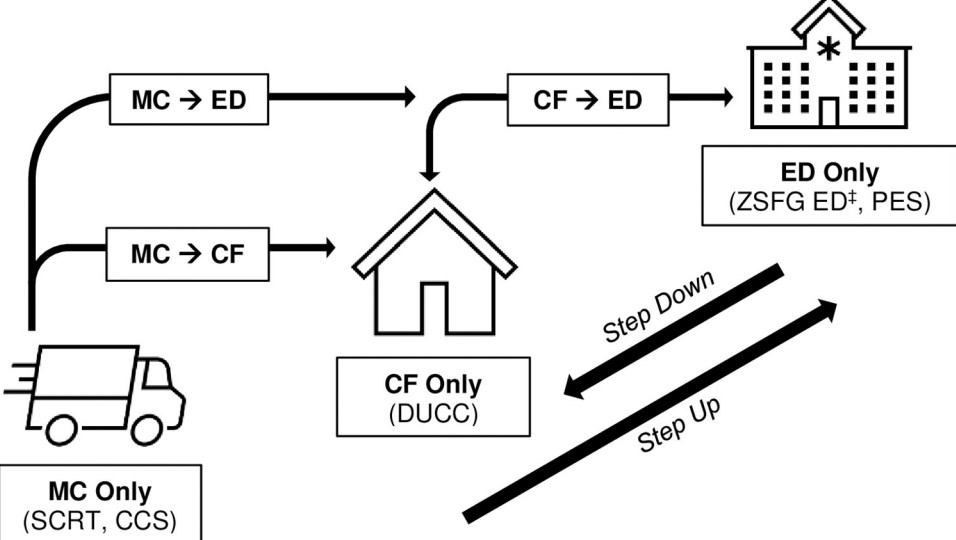

**Fig 2. System map of crisis system entry points in San Francisco used for defining index crisis episodes.** MC–mobile crisis; CF–crisis facility; ED–emergency department; SCRT–Street Crisis Response Team; CCS–Comprehensive Crisis Services (adult/child mobile crisis team in San Francisco); DUCC–DORE Urgent Care Center; ZSFG ED–Zuckerberg San Francisco General Hospital Emergency Department; PES–Psychiatric Emergency Services. ‡—ZSFG ED limited to medical episodes with primary ICD-10 diagnosis of mental health, substance use, or suicide Z-codes.

clinicians use to assess every person who enters the San Francisco County Jail, as well as Epic following their transition to this EHR in October 2021.

We will use a two-step process for linking records from the Avatar, Epic (including HMIS) and JIMS EHRs by first matching on demographic data fields such as first and last name, date of birth, legal sex, and at least one additional element (e.g., Social Security Number (SSN), full street address, phone number, or email address), and then an additional round of name matching using a Jaro-Winkler based process with matching parameter = 1 (i.e., exact match) [20, 21]. Unique individuals and episodes will be assigned anonymized identifiers to create a limited dataset that includes information about demographics, clinical attributes, dates of service, and zip codes. Authors did not have access to personal health information or other personal data that could identify individuals during or after data collection for the ITS.

**Index crisis episode.** Conducting an ITS that includes a time-dependent outcome—in this case a routine care episode, acute care episode, housing assessment, or jail entry within 30 days following a crisis episode—requires defining an index crisis episode to start the clock for the outcome time interval. Furthermore, given that there may be multiple index episodes per unique individual, the index crisis episode needs to be defined per ITS interval, which, based on our preliminary power analysis, will be divided as one calendar month per interval. The index crisis episode will therefore be defined as the first instance in a given month that an individual has a crisis episode, with the 30-day post-crisis outcome period trailing the end date of the index episode.

We also need to account for the fact that a single crisis episode may result in multiple contacts with different settings in the acute behavioral health care system. To account for this variability, we will use our clinical knowledge of common care pathways as the basis for a crisis system map (Fig 2) that defines different sequences of care as either a step-up in service intensity (e.g., mobile crisis followed by crisis stabilization or ED), a step-down in service intensity (e.g., PES followed by crisis stabilization), or a distinct crisis episode that mostly likely signifies

reutilization (e.g., PES followed by mobile crisis, or two consecutive mobile crisis episodes). We will then combine sequences with a step up or down in service intensity that occur within two or fewer days between the end of the first and start of the second clinical setting, so as not to miscount these care transitions as reutilization. This approach also allows us to create a variable describing these crisis system trajectories, thus allowing us to identify potential differences between index crisis episodes based on crisis system entry point and end point. All crisis services utilized three or more days after the index crisis episode end date will be considered separate from the index crisis episode.

**Outcome variables.** We will build ITS models using the following four repeated measures outcomes: 1) post-crisis episode routine care utilization within 30 days; 2) post-crisis episode crisis service reutilization within 30 days; 3) post-crisis episode housing assessment within 30 days; and 4) post-crisis episode jail entry within 30 days. Routine care services will include receipt of outpatient services within 30-days following the index crisis episode in programs related to mental health, substance use, primary care, and integrated behavioral health in primary care. We will exclude residential programs and other services that are not intended to serve as routine care nor as a front-door crisis service. Acute care reutilization will include when an individual has a crisis care episode within 30-days following discharge from the index crisis episode. Housing assessment will be determined among those who are identified as PEH as having an HMIS record of receiving a housing assessment within 30-days following the index crisis episode. Finally, we will measure jail entry within 30-days following the index crisis episode based on clinical records from JIMS or Epic that are documented for every person who enters the San Francisco County Jail.

**Covariates.** Demographic variables for age at time of service, gender identity, sexual orientation, race/ethnicity, housing status and insurance status will be developed in accordance with existing SFDPH reporting guidelines. Gender identity will be determined by information on each person's sex at birth as well as self-reported gender identity to create categories for cisgender male or female, transgender male or female, and genderqueer or nonbinary. Sexual orientation is based on self-report. Race and ethnicity are recorded separately in both EHR systems and will be cross-referenced and then combined into a single variable by replacing race with ethnicity for those who identify as Hispanic/Latinx [22]. SFDPH defines someone as a person experiencing homelessness if they utilize a service that indicates housing instability (e.g., emergency shelter) or self-report homelessness while accessing health care services. Insurance status at the time of an encounter will be based on EHR billing records and grouped into descriptive categories (e.g., private versus public insurance). A location variable using zip code will be based on last location documented prior to the crisis episode. Multiple imputation will be used to account for missing data.

Diagnoses associated with service encounters will be categorized using the primary ICD-10 diagnostic code based on the Health Care Utilization Project's Clinical Classifications Software Refined (CCSR) [23]. Given the unreliability of diagnosis data at the time of a crisis encounter [24, 25], we will use diagnoses made in routine service settings in the 90 days prior to the index crisis episode and, only if routine care is unavailable, will use acute-care settings diagnoses.

Additional clinical variables will include whether the index crisis episode resulted in an involuntary psychiatric hold; suicidality as part of the presentation (based on ICD-10 Z codes as well as clinical documentation or indication for involuntary holds as "danger to self"); and violence risk (based on clinical documentation or indication for involuntary holds as "danger to others") [26]. We will also control for continuous variables describing the number of crisis services, routine care, jail, or housing assessment encounters in the 12-months prior to the index crisis episode. We will also assess the number of crisis services, routine care, jail, or housing assessment encounters in the 12-months prior to the index crisis episode.

**Interrupted time series analysis.** The ITS analysis will examine three periods: 1) Pre-SCRT baseline (November 2019 to November 2020), 2) SCRT partial implementation (December 2020 to July 2021), and 3) SCRT full implementation (August 2021 to August 2022). Each of the outcome measures will be computed as a monthly proportion, with the numerator equaling the number of individuals meeting criteria for each outcome and the denominator equaling the total target population in a given month. Using month-long time intervals would yield between 8 and 12 data points per time period, though the final interval length may change depending on the trade-offs between length of observation and statistical power.

Generalized estimating equations (GEE) will be used, with robust standard errors to account for within-person correlation [27], to analyze the trends in outcomes pre-implementation, during partial implementation, and after the implementation of SCRT. Models will include a variable for time (month) after the beginning of our observation period, a variable for time each time period subsequent to baseline, and the interaction between these variables to assess change in the trajectory of each outcome. Models will be adjusted by the covariates described above and interaction terms constructed to understand the relative effects of covariates and the ITS variables.

This analysis will use several design and analysis strategies to account for potential threats to internal validity. First, the study design using two nonequivalent groups with staggered implementation (crisis episodes with zip codes corresponding to neighborhoods that did or did not have the SCRT active during the partial implementation phase prior to citywide expansion) will allow for a between-site comparison of the pilot catchment area relative to the non-pilot areas (Fig 1). Additional sub-analyses will examine within-site differences for the pilot neighborhoods across the three time periods, and, separately, within-site differences for the non-pilot neighborhoods. Second, a non-equivalent non-treatment control group (non-homeless adults accessing crisis services) will be compared to the target population (homeless adults accessing crisis services) using a difference of differences approach to account for secular variations in mental health and substance use service utilization. Third, a series of non-equivalent dependent outcome variables that are not expected to be impacted by the implementation of the SCRT (e.g., non-crisis initiation of outpatient mental health services) will be evaluated across the same time periods with a similar goal of accounting for secular variations in mental health service utilization. Additional issues such as autocorrelation of repeated measures on individuals will be corrected for in the final analyses.

**Equity analysis.** We will conduct additional ITS analyses of each model by stratifying the population by our covariate on race and ethnicity (defined by patient self-report in EHR demographic records). By stratifying the total population into sub-categories of interest, including ethnoracial groups, and comparing the outcomes of the ITS analyses, we will be able to describe whether the potential impact of the SCRT intervention was equitably distributed across racial groups [28–31]. Furthermore, we will be able to identify if potential baseline disparities are perpetuated or reduced by implementation of the SCRT, as has been described in the RE-AIM model [32, 33].

## Qualitative design

To identify individual, community and societal-level factors associated with optimal and sub-optimal implementation of the SCRT from a client perspective, semi-structured interviews will be conducted with recipients of the SCRT and post-crisis outreach services. Qualitative data will be analyzed using a thematic analysis approach and further support interpretation of the quantitative findings.

**Interview participant recruitment and consent.** Our study team will partner with the post-SCRT outreach clinicians to engage with individuals who received SCRT services to help

recruit a sample purposively selected based on receipt of services and the quantitative post-crisis routine care and reutilization outcomes. Potential participants will be eligible if they received SCRT services 7 to 90 days prior to contact with the study team, to provide time for the resolution of their recent crisis while limiting potential effects of recall bias. During post-crisis follow-up, OCC clinicians will describe the study and, for those interested, document consent for our team to contact them and to review their medical records. Those who consent to be contacted by the research team will have the option to provide their contact information (i.e., phone number, email address) for the research team to identify and contact them for recruitment purposes. Potential participants will also be recruited directly using various methods, such as circulating flyers in areas that frequently serve PEH (e.g., shelters, social service organizations, and on SCRT units themselves). We will then contact prospective participants and arrange to meet participants in person or speak by telephone. Consent will be obtained immediately prior to the interview. Verbal consent procedures were approved by the UCSF Committee on Human Subjects Research to allow for telephonic interviews; written consent was also obtained for in-person interviews. A $60 gift card incentive will be offered to all participants upon completion of the interview. Community stakeholders will be engaged to provide additional suggestions about recruitment strategies.

**Semi-Structured Interviews.** Questions for the semi-structured interviews will be developed *a priori* based on examination of the literature, our team's clinical experience with crisis services, and feedback from the SCRT team and community stakeholders. The 30- to 60-minute interview will include specific questions about baseline engagement in health care and housing systems, SCRT's accessibility, SCRT intervention and assessment, post-crisis linkage to care, and overall client experience. Open-ended questions and follow-up prompts will aim to elicit the participant's perception of the encounter and SCRT's role in their broader experiences of homelessness, mental illness and substance use. All interviews will be conducted by teams of two researchers, audio recorded and transcribed for qualitative analysis using Atlas.ti Version 9 software.

**Qualitative coding and analysis.** The interview findings will be coded for salient themes using a grounded theory approach [34]. An initial codebook will be developed based on the interview guide, prior literature and overall study goals, and then during the analysis will allow for new codes and themes to emerge organically from the text [35]. Our research team will meet weekly about emerging themes and to discuss iterative changes to the codebook until group consensus determines that saturation has been reached [36]. Finally, we will examine codes and discuss possible models that help organize the themes, such as Bronfebrenner's Socio-ecological Model [37].

## Mixed methods analysis

Once both the ITS and qualitative analyses have been completed, the results of each will be examined in conjunction with the other to assist with interpretation of the overall study findings. For example, measuring quantitative service utilization outcomes among interview participants can help inform how we interpret their reports of how SCRT impacted their lives. If the equity analyses produce concerning signals for disparities in care, we could incorporate this information into how we analyze qualitative findings about discrimination. Another example would be if we find a lack of significant change in the ITS analysis of post-crisis routine care utilization after implementation of SCRT, in which case our interpretation may be informed by qualitative descriptions of barriers to accessing routine care. These kinds of mixed methods approaches allow us to leverage the nuanced details of qualitative research and the more representative findings in population-level quantitative research to arrive at a stronger set of interpretations and conclusions.

## Discussion

Many states and counties throughout the U.S. have turned to mobile crisis services as a potential cost-effective solution to constraints in behavioral health service capacity. A 2020 report issued by the Substance Abuse and Mental Health Services Administration, titled "National Guidelines for Crisis Care–A Best Practice Toolkit," lays out essential services for a crisis continuum of care: call centers, mobile teams, and stabilization centers [38]. Programs such as SCRT are key to health systems seeking solutions to divert 9-1-1 calls away from law enforcement and instead to specialized behavioral health clinicians who can triage and link clients to an appropriate level of care. Further implementation science research is essential to grow the evidence base for effective mobile crisis models to help address the behavioral health needs of people experiencing a crisis, especially vulnerable populations such as people experiencing homelessness and health disparities.

The study described in this protocol exemplifies how implementation science methods can increase our understanding of the effectiveness and acceptability of mobile crisis response programs. By describing factors and mechanisms that facilitate or impede the effectiveness of SCRT in diverting clients from unnecessary additional crisis services or jail entry while improving linkage to routine care and housing services, this study will inform future strategies for implementing mobile crisis interventions into other settings. Additionally, the qualitative approaches will provide a nuanced understanding of how an intervention such as SCRT impacts the lives of adults experiencing homelessness in San Francisco and allow for more refined interpretation of the quantitative findings.

### Limitations

There are several limitations to the research methods described above. While ITS designs can measure the impact of a non-randomized intervention, this quasi-experimental approach may not yield definitive results and may be specific to San Francisco and therefore less generalizable to other regions. The electronic records used to describe health, housing and jail service utilization does not capture all services provided in settings outside of the San Francisco Department of Public Health. Though data from 9-1-1 dispatch might help identify which cases are being diverted from SFPD to the SCRT, these data will not be available for this evaluation. It is not possible to fully account for the potential impacts of COVID on data collection and service utilization. Incomplete data on the location of PEH may impede any location-specific analysis; in this case, only the pre-pilot and full implementation segments would be included in the ITS. Limitations in the semi-structured interviews may include sampling bias that results from recruitment through the clinical program, the difficulty of locating potential participants, and the need to exclude potential participants who do not have capacity to consent due to psychiatric or other symptoms; recall bias related to traumatic experiences while in crisis; and desirability bias due to difficulty describing negative experiences of the health care system.

### Conclusion

Studying the implementation of this novel mobile crisis service model will take a step toward increasing our understanding and uncovering mechanisms of impact associated with these widely utilized yet under-evaluated programs. As mobile crisis programs are introduced across the U.S., we aim to provide a model for how mixed-methods evaluations can be achieved in real-world program crisis service settings as a way to advance the literature in this important health services area.

## Supporting information

**S1 Checklist.**
(DOCX)

## Author Contributions

**Conceptualization:** Matthew L. Goldman.

**Writing – original draft:** Deepa Manjanatha.

**Writing – review & editing:** Megan McDaniel, Monica L. Rose, Glenn-Milo Santos, Starley B. Shade, Ann A. Lazar, Janet J. Myers, Margaret A. Handley, Phillip O. Coffin.

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
