## [Decision Letter · Decision Letter 0]

17 Jul 2023

PONE-D-23-12980Impact of San Francisco’s new Street Crisis Response Team on service use among people experiencing homelessness with mental and substance use disorders: A mixed methods study protocolPLOS ONE

Dear Dr. Goldman,

Thank you for submitting your manuscript to PLOS ONE. After careful consideration, we feel that it has merit but does not fully meet PLOS ONE’s publication criteria as it currently stands. Therefore, we invite you to submit a revised version of the manuscript that addresses the points raised during the review process.

We look forward to receiving your revised manuscript.

Kind regards,

De-Chih Lee, Ph.D.

Academic Editor

PLOS ONE

Journal Requirements

“This work was supported by the Robert Wood Johnson Foundation’s Health Systems Transformation Research Coordinating Center Call for Proposals: Research to

Advance Models of Care for Medicaid-Eligible Populations (Grant #78236).”

“Dr. Goldman is a paid research consultant for Vibrant Emotional Health, the National Council for Mental Wellbeing, Peg’s Foundation, the University of California, Davis, and the Research Foundation for Mental Hygiene, Inc.”

6. Please upload a new copy of Figure 1 as the detail is not clear. Please follow the link for more information: https://blogs.plos.org/plos/2019/06/looking-good-tips-for-creating-your-plos-figures-graphics/

7. We note that Figure 1 in your submission contain map images which may be copyrighted. All PLOS content is published under the Creative Commons Attribution License (CC BY 4.0), which means that the manuscript, images, and Supporting Information files will be freely available online, and any third party is permitted to access, download, copy, distribute, and use these materials in any way, even commercially, with proper attribution. For these reasons, we cannot publish previously copyrighted maps or satellite images created using proprietary data, such as Google software (Google Maps, Street View, and Earth). For more information, see our copyright guidelines: http://journals.plos.org/plosone/s/licenses-and-copyright.

Additional Editor Comments:

Please have the author reply and make minor revisions according to the three reviewers' comments.

Reviewers' comments:

Reviewer's Responses to Questions

**Comments to the Author**

1. Does the manuscript provide a valid rationale for the proposed study, with clearly identified and justified research questions?

Reviewer #1: Yes

Reviewer #2: Yes

Reviewer #3: Yes

2. Is the protocol technically sound and planned in a manner that will lead to a meaningful outcome and allow testing the stated hypotheses?

Reviewer #1: Yes

Reviewer #2: Yes

Reviewer #3: Partly

3. Is the methodology feasible and described in sufficient detail to allow the work to be replicable?

Reviewer #1: Yes

Reviewer #2: Yes

Reviewer #3: No

4. Have the authors described where all data underlying the findings will be made available when the study is complete?

Reviewer #1: No

Reviewer #2: Yes

Reviewer #3: Yes

5. Is the manuscript presented in an intelligible fashion and written in standard English?

Reviewer #1: Yes

Reviewer #2: Yes

Reviewer #3: Yes

6. Review Comments to the Author

You may also provide optional suggestions and comments to authors that they might find helpful in planning their study.

Reviewer #1: The protocol described represents an important step in establishing best practices for mobile crisis units and their evaluation.

The authors propose to use mixed methods (quantitative and qualitative approaches) to examine a recently-established mobile crisis unit in San Francisco, California.

Using an interrupted time series analysis, they will measure four main outcomes after an index crisis episode: routine care utilization and housing services assessment (outcomes consistent with stabilization) and crisis care reutilization and jail entry (consistent with de-stabilization). They will compare these outcomes before and after the rollout of the SCRT mobile crisis service in San Francisco, including a pilot period in which the service was rolled out in only some zip codes, providing both time and place-related comparisons. They have appropriately described ways to control for secular patterns in the data.

Using semi-structured interviews, they will obtain qualitative data regarding recipients’ experience with crisis services in the 7 to 90 days following a crisis episode.

The procedures to integrate data from multiple sources are adequately described and seem feasible.

Specific Suggestions:

One potential obstacle to the hoped-for improvements to come from the mobile crisis service and referral plan is a lack of available service providers. For instance, a person could go through the housing assessment but still experience a re-utilization of crisis services because there were not enough housing units, they remained unhoused and a crisis once again ensued on the street. The authors should provide some way of evaluating the availability of such services, perhaps in the qualitative analysis. At the very least, they should comment on it.

Second, in the ITS section (page 12-13), please clarify how the “total target population” will be calculated. Also, how is it estimated that there will be between 8-12 data points per time period?

Third, in the description of the semi-structured interviews, it is unclear whether these have already been performed and remain to be analyzed, or if they will be performed on subjects whose usage of the crisis services does not overlap with the time frame of the qualitative analysis (2020-2022). Please clarify.

Similarly, if the interviews have not yet been performed, will the authors be able to obtain service utilization records for the interview participants as described on page 16, lines 358-359, along with appropriate consent?

Fourth, Figure 1 was very difficult to see, both in the PDF version and when printed.

Finally, please add a statement describing how data will (or will not) be made available after the study is complete.

Reviewer #2: First let me congratulate you on your ambitious and significant study. I enjoyed reading the protocol and think that the methods you are employing could be used in a number of studies to evaluate implementation of real-world interventions that cannot be easily randomized. A few minor comments.

It might be useful to include some hypotheses so that readers can better understand how you expect the outcomes to change. For example, I am assuming that visits to housing services or to primary care would be seen as a positive outcome, while no change or an increase in arrest would be seen negatively.

In describing the equity analysis you write, "Furthermore, we will be able to identify if potential baseline disparities are perpetuated or reduced by implementation of the SCRT, as has been described in the RE-AIM model." Since you have not really described the RE-AIM framework in the paper, it would be good to briefly describe what changes in equity you expect to see (e.g., reach, adoption).

How will eligibility be determined for potential qualitative participants who self-recruit by responding to a flyer. Will you check to see if they have received SCRT services?

When describing methods for matching data you switch unexpectedly to the past tense. Since, I assume that the matching has not yet happened, it would seem better to use the future tense with this as well. In the discussion, "The electronic records used to describe health, housing and jail service utilization does not capture all services provided..." I believe it should be do not capture.

Reviewer #3: This is a significant and worthy study of a critical topic. The QUANT-QUAL implementation methods are appropriate for examining this program launch. Two weaknesses that are correctable:

1. There is a serious imbalance between the detailed quantitative methodology and the sketchy outline provided for the qualitative methodology. In a QUANT-QUAL design, both must be addressed with rigor and (this is missing) a precise plan for staging the merging or integration of the two methods. The protocol falls apart on this dimension. I recommend that the authors consult the NIH guidelines for combining QUANT-QUAL:

https://obssr.od.nih.gov/research-resources/mixed-methods-research

Another helpful resource is:

https://www.ncbi.nlm.nih.gov/pmc/articles/PMC7756351/

Both resources describe a clearly designed plan for bringing data from both approaches together and note that a precise plan about when and how should be described. The second resource describes the basic qualifications for rigor in qualitative methods which are missing from the protocol.

2. While the quantitative analytic design is well-described, the equity analysis suffers from inattention to intersectionality and cumulative stress of multiple social identities. The design did not clearly identify how singular versus multiple identities would be accounted for, i.e. the differences between inequities experienced by a white male with serious mental illness symptoms and a black female with similar symptoms is not addressed. The analysis should fully represent these intersecting factors -- it appears that they are treated as discreet factors, not cumulative.

7. PLOS authors have the option to publish the peer review history of their article (what does this mean?). If published, this will include your full peer review and any attached files.

Reviewer #1: **Yes: **Nicole L. Schramm-Sapyta

Reviewer #2: No

Reviewer #3: No

---

## [Author Response · Author response to Decision Letter 0]

1 Nov 2023

(10/15/23) Dear Dr. De-Chih Lee,

Thank you for the opportunity to submit a revision of this Study Protocol, titled, " Impact of San Francisco’s new Street Crisis Response Team on service use among people experiencing homelessness with mental and substance use disorders: A mixed methods study protocol." Please see below for our responses to the comments about journal requirements and from the reviewers. 

Journal Requirements: 

>>RESPONSE: Thank you for giving us the opportunity to correct the manuscript to adhere to PLOS ONE’s style requirements. We have revised the manuscript formatting so that it complies with the journal’s style requirements.

>>RESPONSE: Thank you for your attention to ethical research among minors. On Page 8 of our Methods section, under the subsection “Study Population,” we state that “the study population will be defined based on age greater than 18 years.” We therefore did not need to obtain consent from parents or guardians and the IRB did not need to waive consent because we excluded minors from our study population. 

“This work was supported by the Robert Wood Johnson Foundation’s Health Systems Transformation Research Coordinating Center Call for Proposals: Research to

Advance Models of Care for Medicaid-Eligible Populations (Grant #78236).”

>>RESPONSE: Thank you for bringing this to our attention. We have updated the funding statement in our manuscript and cover letter to include the statement that you provided, which accurately describes the funder’s role in our study: ”The funders had no role in study design, data collection and analysis, decision to publish, or preparation of the manuscript.”

“Dr. Goldman is a paid research consultant for Vibrant Emotional Health, the National Council for Mental Wellbeing, Peg’s Foundation, the University of California, Davis, and the Research Foundation for Mental Hygiene, Inc.”

>>RESPONSE: We reviewed PLOS ONE policies on sharing data and materials and confirmed that there is no impact. Based on this information, we have added the following statement to the manuscript and cover letter: "This does not alter our adherence to PLOS ONE policies on sharing data and materials.” 

5. PLOS requires an ORCID iD for the corresponding author in Editorial Manager on papers submitted after December 6th, 2016. Please ensure that you have an ORCID iD and that itis validated in Editorial Manager. To do this, go to ‘Update my Information’ (in the upper left-hand corner of the main menu), and click on the Fetch/Validate link next to the ORCID field. This will take you to the ORCID site and allow you to create a new iD or authenticate a pre-existing iD in Editorial Manager. Please see the following video for instructions on linking an ORCID iD to your Editorial Manager account: https://www.youtube.com/watch?v=_xcclfuvtxQ

>>RESPONSE: The ORCID ID for the corresponding author, Matthew L. Goldman, is: 0000-0002-2252-9285

6. Please upload a new copy of Figure 1 as the detail is not clear. Please follow the link for more information: https://blogs.plos.org/plos/2019/06/looking-good-tips-for-creating-your-plos-figures-graphics/

>>RESPONSE: We have recreated Figure 1 to improve clarity and to abide by the guidance in comment 7 regarding using an uncopyrighted map image (USGS National Map Viewer). 

7. We note that Figure 1 in your submission contain map images which may be copyrighted. All PLOS content is published under the Creative Commons Attribution License (CC BY 4.0), which means that the manuscript, images, and Supporting Information files will be freely available online, and any third party is permitted to access, download, copy, distribute, and use these materials in any way, even commercially, with proper attribution. For these reasons, we cannot publish previously copyrighted maps or satellite images created using proprietary data, such as Google software (Google Maps, StreetView, and Earth). For more information, see our copyright guidelines:http://journals.plos.org/plosone/s/licenses-and-copyright.

“I request permission for the open-access journal PLOS ONE to publish XXX under the Creative Commons Attribution License (CCAL) CC BY 4.0(http://creativecommons.org/licenses/by/4.0/). Please be aware that this license allows unrestricted use and distribution, even commercially, by third parties. Please reply and provide explicit written permission to publish XXX under a CC BY license and complete the attached form.”

USGS EROS (Earth Resources Observatory and Science (EROS) Center) (public domain):http://eros.usgs.gov/#

>>RESPONSE: We have provided a new version of the map image in figure 1 using the USGS National Map Viewer that does not require copyright. 

>>RESPONSE: We have reviewed our reference list and confirmed that it is complete and correct, and no retracted papers were included in the list. 

Reviewer #1

9. The protocol described represents an important step in establishing best practices for mobile crisis units and their evaluation. The authors propose to use mixed methods (quantitative and qualitative approaches) to examine a recently-established mobile crisis unit in San Francisco, California. Using an interrupted time series analysis, they will measure four main outcomes after an index crisis episode: routine care utilization and housing services assessment (outcomes consistent with stabilization) and crisis care reutilization and jail entry (consistent with de-stabilization). They will compare these outcomes before and after the rollout of the SCRT mobile crisis service in San Francisco, including a pilot period in which the service was rolled out in only some zip codes, providing both time and place-related comparisons. They have appropriately described ways to control for secular patterns in the data. Using semi-structured interviews, they will obtain qualitative data regarding recipients’ experience with crisis services in the 7 to 90 days following a crisis episode. The procedures to integrate data from multiple sources are adequately described and seem feasible. Specific Suggestions: 

>>RESPONSE: Thank you for your review of the manuscript. 

10. (1) One potential obstacle to the hoped-for improvements to come from the mobile crisis service and referral plan is a lack of available service providers. For instance, a person could go through the housing assessment but still experience a re-utilization of crisis services because there were not enough housing units, they remained unhoused and a crisis once again ensued on the street. The authors should provide some way of evaluating the availability of such services, perhaps in the qualitative analysis. At the very least, they should comment on it.

>>RESPONSE: Thank you for this comment about the importance of capturing the lack of available service providers. The qualitative interviews will ask about post-crisis service linkage, which may help inform their experience of difficulty accessing services that were lacking—we have added a clause to the manuscript in the subsection on “Semi-Structured Interviews” that now says, “The 30- to 60-minute interview will include specific questions about baseline engagement in health care and housing systems, SCRT’s accessibility, SCRT intervention and assessment, post-crisis linkage to care and difficulties accessing services due to lack of available resources, and overall client experience.” With regard to the quantitative analysis, we would not have data on provider or service availability outside of the service utilization data, so we added the following sentence to the Limitations section of the Discussion: “Additionally, analyses using electronic health records do not capture the availability of services at the time of post-crisis linkage and are unable to determine whether post-crisis outcomes were impacted by service systems capacity.”

11. (2) Second, in the ITS section (page 12-13), please clarify how the “total target population” will be calculated. 

>>RESPONSE: The total target population refers to the inclusion criteria described in the “Study Population” section: “Therefore, the study population will be defined based on age greater than 18 years, meeting criteria for homelessness in the 12 months prior to or 3 months following the crisis episode, and receipt of crisis care from any of San Francisco’s “front door” programs for people in an acute behavioral health crisis. These settings include two mobile crisis programs (Comprehensive Crisis Services and the SCRT), a crisis stabilization unit (DORE Urgent Care Clinic), and emergency psychiatric services (Zuckerberg San Francisco General Hospital’s Psychiatric Emergency Services [PES] and Emergency Department visits with a primary behavioral health diagnosis) (Fig 2).” To clarify the description of the ITS methods, we have edited the phrase total target population to instead say, “the denominator equaling the total study population in a given month as described above.” 

12. Also, how is it estimated that there will be between 8-12 data points per time period?

>>RESPONSE: Preliminary estimates of the study population, based on historic utilization data from select clinical settings, suggest that there are on the order of a few hundred monthly encounters in the target population. Our statistical analysis suggests that this size population should be powered sufficiently to use month-long time intervals for the interrupted time series, and based on the duration of the three time periods (13 months of Pre-SCRT baseline, 8 months of SCRT partial implementation, and 13 months of SCRT full implementation), this would yield between 8 and 12 data points per time period, which is usually adequate for an ITS analysis. To clarify this in the text, we have added a phrase to the sentence that this is based on preliminary data: “Using month-long time intervals would yield between 8 and 12 data points per time period based on preliminary estimates of the study population size, though the final interval length may change depending on the trade-offs between length of observation and statistical power.”

13. (3) Third, in the description of the semi-structured interviews, it is unclear whether these have already been performed and remain to be analyzed, or if they will be performed on subjects whose usage of the crisis services does not overlap with the time frame of the qualitative analysis (2020-2022). Please clarify. Similarly, if the interviews have not yet been performed, will the authors be able to obtain service utilization records for the interview participants as described on page 16, lines 358-359, along with appropriate consent?

>>RESPONSE: At this point, all interviews have been completed and are in the process of being analyzed, but we would like to keep the voice of this study protocol in the future tense for the sake of consistency and to be referenced in future studies that may draw on these methods. We have therefore opted not to change the language in the methods accordingly but welcome feedback from the journal if we should adjust this approach. With regard to when the interviews were conducted, they were all completed with individuals whose crisis service usage does overlap with the time frame of the quantitative analysis, so the text in the manuscript describing eligibility requirements for the interviews remains accurate: “Potential participants will be eligible if they received SCRT services 7 to 90 days prior to contact with the study team.” 

14. (4) Fourth, Figure 1 was very difficult to see, both in the PDF version and when printed.

>>RESPONSE: We have recreated Figure 1 to improve clarity and to abide by the journal’s guidance regarding using an uncopyrighted map image. 

15. (5) Finally, please add a statement describing how data will (or will not) be made available after the study is complete.

>>RESPONSE: Data acquired for the proposed study will be available upon request. We added the following statement to the Data Availability in the manuscript: “Datasets used and analyzed for the proposed study will be available after study completion from the corresponding author on reasonable request.”

Reviewer #2

16. First let me congratulate you on your ambitious and significant study. I enjoyed reading the protocol and think that the methods you are employing could be used in a number of studies to evaluate implementation of real-world interventions that cannot be easily randomized. A few minor comments. 

>>RESPONSE: Thank you for your review and for the opportunity to respond to your comments. 

17. (1) It might be useful to include some hypotheses so that readers can better understand how you expect the outcomes to change. For example, I am assuming that visits to housing services or to primary care would be seen as a positive outcome, while no change or an increase in arrest would be seen negatively. 

>>RESPONSE: Thank you for this excellent suggestion. We have added the following description of our main hypotheses to the section titled “Interrupted Time Series Analysis”: 

These methods will test the following hypotheses: 

1. Monthly rates of post-crisis episode routine care utilization within 30 days of an index crisis episode will increase following SCRT implementation. 

2. Monthly rates of post-crisis episode crisis service reutilization within 30 days of an index crisis episode will decrease following SCRT implementation. 

3. Monthly rates of post-crisis episode housing assessment within 30 days of an index crisis episode will increase following SCRT implementation. 

4. Monthly rates of post-crisis episode jail entry within 30 days of an index crisis episode will decrease following SCRT implementation. 

18. (2) In describing the equity analysis you write, "Furthermore, we will be able to identify if potential baseline disparities are perpetuated or reduced by implementation of the SCRT, as has been described in the RE-AIM model." Since you have not really described the RE-AIM framework in the paper, it would be good to briefly describe what changes in equity you expect to see (e.g., reach, adoption).

>>RESPONSE: Thank you for catching this. The RE-AIM framework is not essential to this explanation of our disparities analysis, so we have decided to remove the confusing phrase “as has been described in the RE-AIM model” rather than add potentially confusing or extraneous details about that model. 

19. (3) How will eligibility be determined for potential qualitative participants who self-recruit by responding to a flyer. Will you check to see if they have received SCRT services?

>>RESPONSE: Yes, we will confirm that all potential interview participants who self-recruit meet eligibility criteria. We have added the following statement to clarify this: “For those who self-recruit by contacting the study team, we will obtain verbal consent to access electronic health records and the study team will confirm eligibility for the qualitative interviews (i.e., receipt of SCRT services in the previous 7 to 90 days).”

20. (4) When describing methods for matching data you switch unexpectedly to the past tense. Since, I assume that the matching has not yet happened, it would seem better to use the future tense with this as well. In the discussion, "The electronic records used to describe health, housing and jail service utilization does not capture all services provided..." I believe it should be do not capture.

>>RESPONSE: Thank you for bringing this to our attention. We have reviewed and updated these sections to future tense. 

Reviewer #3

21. This is a significant and worthy study of a critical topic. The QUANT-QUAL implementation methods are appropriate for examining this program launch. Two weaknesses that are correctable:

>>RESPONSE: Thank you for reviewing our manuscript and for the opportunity to respond to your comments.

22. (1) There is a serious imbalance between the detailed quantitative methodology and the sketchy outline provided for the qualitative methodology. In a QUANT-QUAL design, both must be addressed with rigor and (this is missing) a precise plan for staging the merging or integration of the two methods. The protocol falls apart on this dimension. I recommend that the authors consult the NIH guidelines for combining QUANT-QUAL: 

https://obssr.od.nih.gov/research-resources/mixed-methods-research

Another helpful resource is:

https://www.ncbi.nlm.nih.gov/pmc/articles/PMC7756351/

Both resources describe a clearly designed plan for bringing data from both approaches together and note that a precise plan about when and how should be described. The second resource describes the basic qualifications for rigor in qualitative methods which are missing from the protocol.

>>RESPONSE: Thank you for this comment requesting further detail about our qualitative methods as well as the QUANT-QUAL approach. We have added additional detail in the “Qualitative Design” section, including: 

For the “Interview Participant Recruitment and Consent” subsection: 

“For those who self-recruit by contacting the study team, we will obtain verbal consent to access electronic health records and the study team will confirm eligibility for the qualitative interviews (i.e., receipt of SCRT services in the previous 7 to 90 days).”

“Finally, we will obtain demographic and service use information from electronic health records for all respondents.”

For the “Semi-structured Interviews” subsection: 

“The 30- to 60-minute interview will include specific questions about baseline engagement in health care and housing systems, SCRT’s accessibility, SCRT intervention and assessment, post-crisis linkage to care and difficulties accessing services due to lack of available resources, and overall client experience.”

“Subsequent iterations of the semi-structured interview guide will be revised periodically during data collection by incorporating emerging themes from early responses. This periodic revision of the interview guide will include sharing the guide with subject matter experts, community stakeholders, and people with lived experience of receiving crisis services to ensure the questions capture important themes for this population. Finally, we will develop an interview team that is diverse in terms of cultural and educational backgrounds to foster reflexivity during data collection.”

For the “Qualitative Coding and Analysis” subsection:

“The codebook will be modified using an inductive process to capture more detailed descriptions of the data. At least two researchers will code each transcript and discuss salient themes based on Braun and Clarke’s “theoretical” thematic analysis.”

“Once the study team codes all transcripts, one of the two original coders will use the final codebook iteration to review their previously coded transcripts to ensure all potentially relevant themes are captured and will resolved any remaining discrepancies.”

Regarding QUANT-QUAL methods, the following statements have been added to the “Mixed Method Analysis” section:

“Guided by Creswell and Plano Clark’s best practices for mixed methods research, we will merge the two types of analyses to mitigate the limitations of the different data types while also highlighting their respective strengths [38]. The ITS analysis is expected to show high-level trends in service usage for SCRT clients but lacks to show any underlying explanation for these trends. On the other hand, the qualitative analysis is expected to show the detailed experiences of an SCRT client, but this information is collected from a subset of SCRT’s clientele and will only generalize to certain circumstances. Any system-level impact of SCRT on service usage among crisis service users will benefit from individual accounts of the program’s implementation and provide a more complete understanding of this crisis service model.

The variety of quantitative and qualitative data that will be collected allows several opportunities to merge the two analyses for better understanding the impact of the SCRT intervention.”

23. (2) While the quantitative analytic design is well-described, the equity analysis suffers from inattention to intersectionality and cumulative stress of multiple social identities. The design did not clearly identify how singular versus multiple identities would be accounted for, i.e. the differences between inequities experienced by a white male with serious mental illness symptoms and a black female with similar symptoms is not addressed. The analysis should fully represent these intersecting factors -- it appears that they are treated as discreet factors, not cumulative.

>>RESPONSE: Thank you for this excellent comment. We agree that it is important to assess for intersectionality in disparities and have added to the following sentence to the Equity Analysis section: “Additional regression analyses using both the stratified and overall samples will use interaction terms to assess for disparities related to intersectionality between other demographic characteristics with race and ethnicity.” 

We appreciate your time and consideration.

Sincerely,

Matthew L. Goldman, M.D., M.S., on behalf of the co-authors

---

## [Decision Letter · Decision Letter 1]

17 Nov 2023

Impact of San Francisco’s new Street Crisis Response Team on service use among people experiencing homelessness with mental and substance use disorders: A mixed methods study protocol

PONE-D-23-12980R1

Dear Dr. Goldman,

We’re pleased to inform you that your manuscript has been judged scientifically suitable for publication and will be formally accepted for publication once it meets all outstanding technical requirements.

Kind regards,

De-Chih Lee, Ph.D.

Academic Editor

PLOS ONE

Additional Editor Comments (optional):

Reviewers' comments:

Reviewer's Responses to Questions

**Comments to the Author**

1. Does the manuscript provide a valid rationale for the proposed study, with clearly identified and justified research questions?

Reviewer #1: Yes

Reviewer #2: Yes

Reviewer #3: Yes

2. Is the protocol technically sound and planned in a manner that will lead to a meaningful outcome and allow testing the stated hypotheses?

Reviewer #1: Yes

Reviewer #2: Yes

Reviewer #3: Yes

3. Is the methodology feasible and described in sufficient detail to allow the work to be replicable?

Reviewer #1: Yes

Reviewer #2: Yes

Reviewer #3: Yes

4. Have the authors described where all data underlying the findings will be made available when the study is complete?

Reviewer #1: Yes

Reviewer #2: Yes

Reviewer #3: Yes

5. Is the manuscript presented in an intelligible fashion and written in standard English?

Reviewer #1: Yes

Reviewer #2: Yes

Reviewer #3: Yes

6. Review Comments to the Author

You may also provide optional suggestions and comments to authors that they might find helpful in planning their study.

Reviewer #1: My only comment remaining is about the use of future tense voice. It seems appropriate to use past-tense when referring to recruitment of subjects for the qualitative analysis (subjects were recruited; subjects were interviewed) and future tense when referring to the analysis (transcripts will be analyzed, etc.). However, i defer to the editor on this matter and would not delay the publication of this important plan based on that issue.

This is very important work and I look forward to the results!

Reviewer #2: Your version with untracked changes does not incorporate the changes found in the tracked changes version. I don't know if the original was included with the resubmission, but it caused some confusion. The second version with tracked changes has the changes described in the authors' response.

Reviewer #3: The authors have been responsive to the reviewers' and editors' requests. Refreshing! Thank you! The responses are thoughtful and have received full responses.

7. PLOS authors have the option to publish the peer review history of their article (what does this mean?). If published, this will include your full peer review and any attached files.

Reviewer #1: **Yes: **Nicole L. Schramm-Sapyta

Reviewer #2: No

Reviewer #3: **Yes: **Linda S. Beeber

---

## [Editor Report · Acceptance letter]

24 Nov 2023

PONE-D-23-12980R1 

Impact of San Francisco’s New Street Crisis Response Team on Service Use Among People Experiencing Homelessness with Mental and Substance Use Disorders: A Mixed Methods Study Protocol 

Dear Dr. Goldman:

I'm pleased to inform you that your manuscript has been deemed suitable for publication in PLOS ONE. Congratulations! Your manuscript is now with our production department. 

Kind regards, 

on behalf of

Dr. De-Chih Lee 

Academic Editor

PLOS ONE